# Effect of Frequency of Exercise on Cognitive Function in Older Adults: Serial Mediation of Depression and Quality of Sleep

**DOI:** 10.3390/ijerph17030709

**Published:** 2020-01-22

**Authors:** Manqiong Yuan, Hanhan Fu, Ruoyun Liu, Ya Fang

**Affiliations:** 1State Key Laboratory of Molecular Vaccinology and Molecular Diagnostics, School of Public Health, Xiamen University, 361102 Xiamen, China; yuanmanqiong@163.com; 2Key Laboratory of Health Technology Assessment of Fujian Province, School of Public Health, Xiamen University, 361102, Xiamen, China; kelly_0701@outlook.com; 3School of Public Health, Xiamen University, 361102, Xiamen, China; ruoyun_l@163.com

**Keywords:** effect of frequency, depression, quality of sleep, cognitive function, serial mediation

## Abstract

Background: Sleep quality and depression are two reciprocal causation socioemotional factors and their roles in the relationship between physical exercise and cognition are still unclear. Methods: A face-to-face survey of 3230 older adults aged 60+ was conducted in Xiamen, China, in 2016. Frequency of exercise (FOE) referred to the number of days of exercise per week. Quality of sleep (QOS) was categorized into five levels: very poor/poor/fair/good/excellent. The 15-item Geriatric Depression Scale (GDS-15) and the Montreal Cognitive Assessment (MoCA) were used to measure depression (DEP) and cognitive function (CF), respectively. Serial multiple mediator models were used. All mediation analyses were analyzed using the SPSS PROCESS macro. Results: 2469 respondents had valid data with mean scores for GDS-15 and MoCA being 1.87 and 21.61, respectively. The direct path from FOE to CF was significant (c’= 0.20, *p* < 0.001). A higher FOE was associated with better QOS (B = 0.04, *p* < 0.01), which in turn was associated with fewer symptoms of DEP (B = −0.40, *p* < 0.001), and further contributed to better CF (B = −0.24, *p* < 0.001). Similarly, a higher FOE was associated with lower GDS-15 scores (B = −0.17, *p* < 0.001) which then resulted in higher MoCA scores (B = −0.24, p < 0.001). However, QOS alone did not alter the relationship between FOE and CF. Conclusions: FOE is a protective factor of CF in older adults. Moreover, CF is influenced by QOS through DEP, without which the working path may disappear.

## 1. Introduction

### 1.1. Physical Activity as a Positive Lifestyle for Cognitive Function

Previous studies have strongly supported the hypothesis that being physically active is associated with a reduced risk of cognitive problems and improved cognitive functioning [1,2,3], specifically in the domains of memory, processing speed, and visuospatial functioning [4,5]. In the UK, a recent longitudinal study of the elderly (N = 11,391, age ≥50) demonstrated that engagement in mild physical activity can reduce the risk of suffering a decline in cognition by 34%–50% during the 8–10 year follow-up period [6]. Emilie et al. [7] used research visits to identify the effect of physical activity in different stages of life on later-life cognitive function and found only activities in old age related to reduced odds of cognitive impairment (OR: 0.77; 95% CI: 0.60–0.99). It has been established in recent years that the mechanisms underlying this relationship can be associated with multiple physiological factors, such as heart rate [8], white matter integrity [9], and hippocampal volume [10]. 

### 1.2. Mental Health and Sleep Quality as Mediators

Physical activity also has an effect on a number of socioemotional factors that further contribute to cognition. A recent study determined that social support and cognitive engagement mediated the effects of physical activities on cognition for the oldest individuals [4]. In addition, increased physical activity was associated with a significant improvement in depression and anxiety [11,12]. Furthermore, mild physical activity and moderate to vigorous activity can reduce the risk of depressive symptoms by 13% and 19%, respectively [13]. The duration of exercise each day was also a protective factor for postnatal depressive symptoms [14]. Low mood was linked to poorer performance [15,16]. A decline in cognition caused by low mood principally involved memory, executive function, and attention, often affected by physical activity [17]. 

Generally speaking, sleep deprivation has negative consequences on cognition. Both time of awakening and rapid eye movement have proven to be important in a variety of cognitive functions, especially executive function [18]. There is also a growing body of literature that has recognized that physical exercise is a critical contributor to quality of sleep. For example, acute exercise was shown to be beneficial to total sleep duration [19,20] in addition to assisting in reducing sleep disturbance [21]. Given the relationship between sleep and cognitive performance and between sleep and physical activity, quality of sleep might be considered a potential mechanism for physical activity to improve cognition. However, few studies have included all three variables in one model. Research aimed at better understanding this complex interplay is critically required. 

### 1.3. A Mutual Relationship between Depression and Sleep Quality 

Previous research has suggested that there are complex and bidirectional relationships between sleep and depression. For example, a cross-sectional study by Kaneita [22] described a linear inverse trend between sleep sufficiency and symptoms of depression, demonstrating that the higher the effectiveness of sleep, the lower the symptoms of depression. Similarly, a U-shaped link between the two factors above has recently been proven, the results also indicating that depressive symptoms shorten an individual’s duration of sleep [23]. Data from a number of studies have suggested that inadequate sleep is both a common symptom and risk factor for psychological disorders such as anxiety and depression [24].

Although a number of studies have explored the independent contributions of depression and quality of sleep on the relationship between physical activity and cognitive function, no studies have explored all these factors in combination. Investigating the mediating relationships between them may contribute to the elucidation of the relevant mechanisms. Given that depression and sleep are two underlying mediators of the way physical activity affects cognitive functioning, and the relationship between the two is mutual, we will first establish the roles of two mediating factors in this study. Subsequently, two multi-mediation models were constructed linking frequency of exercise, quality of sleep, depression, and cognitive function, exploring the bidirectional relationships between quality of sleep and depression, and accessing a specific mediating working path. 

## 2. Materials and Methods

### 2.1. Participants

In 2016, a survey of elderly people older than 60 years of age was conducted in Xiamen. A multi-stage sampling technique was used for data collection. Within all six districts of Xiamen, a sample of 16 sub-districts was included. For each sub-district, one-fifth of the community was selected. In total, 44 communities were selected. Finally, eligible individuals were randomly selected from each community based on the proportion of eligible subjects, resulting in 3230 valid questionnaires being obtained. The ethics committee of the School of Public Health, Xiamen University approved the study. All participants gave written informed consent prior to data collection.

### 2.2. Instruments

Frequency of exercise was obtained with the question “How many days a week do you work out?” Frequency was converted into a score based on the following five levels: 0 = never; 1 = 1–2 days per week; 2 = 3–4 days per week; 3 = 5–6 days per week; 4 = every day. Furthermore, the exercise in this study was limited to the physical activity with the purpose of improving health instead of work and housework and lasting more than 20 minutes. Each individual’s quality of sleep was measured with the single validated item: ”How do you rate your quality of sleep?“ (0 = very poor, 1 = poor, 2 = fair, 3 = good, 4 = very good). The subject’s self-reported health status was assessed by asking “Compared to your peers, in general, how do you feel about your health?” with five options: very poor, poor, fair, good, and very good. Prevalence of hypertension and diabetes were assessed by asking respondents “Did you have hypertension which was confirmed by the doctor?” and “Did you have diabetes which has been diagnosed in the hospital already?”, respectively. Days of drinking tea per week were measured by answers to the question “How many days per week do you drink tea?”

The 15-item Geriatric Depression Scale (GDS-15), which can be completed in 5–7 minutes, was used to measure depression. GDS-15 has been validated as a screening tool for the elderly, with a reported sensitivity of between 79% and 100% and specificity between 67% and 82% [25,26]. It consists of 15 items with self-reported measures about feelings of daily life. The answers were labeled as yes (1) or no (0). Positive items were reverse coded (e.g., “Are you satisfied with your life?”). Total scores ranged from 0 to 15, with higher scores indicating greater depressive symptoms.

The Montreal Cognitive Assessment (MoCA) was used for cognitive screening, a test with a high level of inter-rater reliability and internal consistency for measuring cognitive function. It can be completed in approximately 20 minutes [27]. To adjust for educational measurement bias, a value of 2 was added to the final MoCA score for participants with less than 6 years of education, and one point for participants with 7–12 years of education [28]. Higher scores reflected better cognitive function. A bespoke questionnaire was developed to determine age, gender, residence, level of education, quality of sleep, and frequency of exercise.

### 2.3. Statistical Analysis

Descriptive statistics and Pearson correlation coefficients were used to measure the degree of association between two variables in the data. Statistical significance of the effects of mediation was examined through the plug-in application PROCESS developed by Hayes (2013), an approach based on an ordinary least squares regression model and the bootstrap method. Bootstrap analyses were conducted according to the guidelines provided by Hayes, using the IBM SPSS PROCESS (written by Andrew F. Hayes) macro running Serial Multiple Mediation-Model 6 [29]. The statistical significance of the mediating variable was examined over 10,000 bootstrap samples. This method generated an estimate of the indirect effect, including 95% confidence intervals. When zero was not within the 95% confidence limits, one should conclude that the indirect effect was significantly different from zero at *p* < 0.05; thus, the effect of the independent variable on the dependent variable was mediated by the proposed mediating variable. The significance level in the current research was set as 0.05. IBM SPSS v23.0 software (SPSS Inc., Chicago, IL, USA) was used to analyze the research data.

In all analyses, adjustments for potential confounding variables, such as age, gender (dummy parameterized, male = 0, female = 1), and years of education were included. 

## 3. Results

### 3.1. Descriptive

In order to avoid an adverse pathway between frequency of exercise and cognitive function, participants with extremely low MoCA scores were excluded, with recommended thresholds of 11, 14, and 16 for ≤5, 6–8, and ≥9 years of education, respectively [30]. A total of 2469 participants completed all entries on the questionnaire, and those participants whose questionnaires were incomplete were excluded (n = 431, 13.34%). Of the respondents, 53.4% were males, and less than half of the participants lived in rural areas. Only 10.9% of the elderly considered themselves to be in poorer health, and the proportion of people with hypertension and diabetes were 34.5% and 10.4%, respectively. Most seniors (65.3%) reported drinking tea 5–6 days a week. People who exercised every day of the week comprised the largest proportion, at 45.7%. Mean age ± S.D. at the time of the interview was 69.23 ± 7.14 years. Mean years of education was 5.77. The mean Total MoCA and GDS-15 scores were 21.61 ± 4.92 (range 0–30) and 1.87 ± 2.22 (range 0–15), respectively. The findings of the Pearson correlation tests are presented in Table 1, which indicated that a positive significant relationship was found between the frequency of exercise and quality of sleep. Moreover, they were also both correlated with depression. Broadly speaking, we found that there were positive significant relationships between exercise frequency and sleep quality with cognitive function, but a negative significant relationship between GDS-15 score and cognitive function.

### 3.2. Serial Multiple Mediational Analyses

Figure 1A,B demonstrates the findings of the two tested models of the mediation roles of depression and quality of sleep in the relationship between frequency of exercise and cognitive function. As can be seen, both the total effect (*c* = 0.24, *SE* = 0.04, t = 5.45, *p* < 0.001) and the direct effect (*c* = 0.20, *SE* = 0.04, t = 4.47, *p* < 0.001) of frequency of exercise on cognitive function were at significant levels, indicating that physical exercise is a protective factor for cognition which therefore suggests partial mediation. As shown in Figure 1A, the direct effect of frequency of exercise on GDS (*B* = −0.17, *SE* = 0.02, t = −7.16, *p* < 0.001) was significant, suggesting that participants who exercised more frequently tended to have fewer symptoms of depression. In addition, the GDS score was negatively associated with cognition (*B* = –0.24, *SE* = 0.04, t = −6.60, *p* < 0.001). However, our results also demonstrated that quality of sleep alone did not mediate the relationships mentioned above (*B* = 0.02, *SE* = 0.01, t = 1.87, *p* > 0.05; *B* = 0.01, *SE* = 0.08, t = 0.18, *p* > 0.05). Although the direct effect of depression as the first mediating variable on the second mediating variable of quality of sleep (*B* = −0.09, *SE* = 0.01, t = −9.66, *p* < 0.001) was statistically significant, the direct effect of quality of sleep on cognitive function was not significant (*B* = 0.01, *SE* = 0.08, t = 0.18, *p* > 0.05). Therefore, frequency of exercise did not have an impact on cognitive function through its influence on depression and quality of sleep serially. Based on this result, depression—the sole mediator—was observed to mediate between frequency of exercise and cognitive function. 

As can be seen in Figure 1B, frequency of exercise was associated with better quality of sleep (*B* = 0.04, *SE* = 0.01, t = 3.22, *p* < 0.001), which in turn was associated with fewer depressive symptoms (*B* = −0.40, *SE* = 0.04, t = −9.66, *p* < 0.001), finally contributing to better cognitive function (*B* = −0.24, *SE* = 0.04, t = −6.60, *p* < 0.001). Similarly, frequency of exercise had a significant negative effect on the GDS-15 score (*B* = −0.16, *SE* = 0.02, t = −6.65, *p* < 0.001) which then negatively affected participants’ cognition (*B* = −0.24, *SE* = 0.04, t = −6.60, *p* < 0.001). 

The statistical significance of indirect effects in the tested model was examined through 10,000 bootstrap samples. Estimates were taken within a 95% confidence interval and bias-corrected. Accelerated results are presented in Table 2 and Table 3. They include the comparisons of indirect effects and specific indirect effects of frequency of exercise through depression and quality of sleep on cognitive function. Contrasting findings presented in pairs between the paths FOE–QOS–DEP–CF and FOE–DEP–CF were included in the current study to determine whether specific indirect effects of mediating variables were stronger than others in model 2. As seen in Table 3, based on a bias-corrected and accelerated (BCa) confidence interval of 95%, within a statistically significant comparison and outside the point estimate interval, the mediation role of depression alone was found to be stronger than the serial multiple mediation role of sleep quality and depression together.

Regrettably, the action pathway between depression and quality of sleep remained unclear in our study. In model 1, the GDS-15 scores showed a significant positive correlation with sleep quality (*B* = −0.10, *SE* = 0.01, t =−9.66, *p* < 0.001), suggesting severe depression may result in worse quality of sleep. Consistently, results from model 2 implied that individuals with a better quality of sleep are more likely to have lower levels of depression (*B* = −0.40, SE = 0.04, t = −9.66, *p* < 0.001). 

## 4. Discussion

This study examined the serial multiple mediation roles of depression and quality of sleep on the relationship between frequency of exercise and cognitive function. The results demonstrated that the serial multiple mediation of quality of sleep and depression and the separate mediation of single mediating variables were statistically significant in the relationship between frequency of exercise and cognitive function. Based on the contrasting pairs of two significant indirect effects, the sizes of the mediating effect of sleep quality and depression was found to differ statistically from depression alone in relation to frequency of exercise and cognitive function, and the effect value of the former, indirect path was larger than that of the latter. 

Among the more significant findings to emerge from this study was the observation that frequency of exercise can affect cognitive performance in the elderly by affecting their quality of sleep, which in turn can influence depression. Prior studies have noted the importance of regular physical activity for quality of sleep, suggesting that individuals with regular exercise training on average experienced a significantly higher total duration of sleep [31] and better quality of sleep [32,33] than their counterparts. Furthermore, our results demonstrate that poor quality of sleep may increase the risk of developing depression. These findings are in accordance with several previous studies [34,35,36] that confirmed the manner in which sleep quality can affect psychological distress in various populations. Surprisingly, we also found that poor mental health can trigger sleep disorders. This combination of findings provides support for the conceptual premise that poor sleep and poor mental health can each increase the risk of the other’s morbidity, consistent with a longitudinal study performed in Japanese adolescents in 2008 [37]. 

Regarding these findings, it can be concluded that frequency of exercise contributes to depression, and thus, cognitive function. It has been proven that physical exercise can alleviate depressive symptoms [38] and anxiety disorders [39]. A possible explanation for this might be that individuals who exercise at least twice a week show a higher sense of coherence and a stronger sense of social integration than those who exercise less frequently or not at all [40], with the social interaction provided by exercise accounting for a large proportion of the effects of depression. Another possible mechanism was discussed by Paluska and Schwenk [41], who proposed that participation in exercise activities temporarily shifts attention away from uncomfortable physical and emotional experiences, resulting in improved effects, a phenomenon termed “distraction hypothesis”.

Surprisingly, when controlled for depression, no significant relationship was found between quality of sleep and cognitive function. This outcome is contrary to that of Falck et al. (2018), who found that quality of sleep had an important role in predicting cognitive function [42]. It is difficult to explain this result, but it might be partly explained by quality of sleep only relating to specific cognitive tasks, such as working memory and abstraction, instead of overall cognitive performance [43]. Therefore, the effect of sleep deprivation on cognitive performance was not apparent in this study.

## 5. Conclusions

To the best of our knowledge, the present study is one of the largest studies of the mechanism of mediation between frequency of exercise and cognitive function in a Chinese population. For the first time, frequency of exercise, quality of sleep, depression, and cognitive function have been included in the same model. Although the mechanism between depression and quality of sleep was not figured out, even after conducting two mediating models, the present findings might help provide the framework for a new way of understanding the relationship in older populations. Moreover, we have identified that compared with depression, quality of sleep appears less important in the underlying intermediary mechanism. Based on the general result of the current research, milder depression and better quality of sleep have been determined to function as a buffer against cognitive decline. In order to reduce seniors’ depressive symptoms and increase their quality of sleep and cognitive function, more physical activity programs like dancing and Baduanjing exercise could be recommended in communities. Taking a walk, which is also considered as a convenient way to exercise, may be organized for different groups. Health education concerning the benefits of physical exercise on depression, sleep quality, and cognitive function can be conducted to raise relevant issues. The current study also has some limitations. Firstly, cross-sectional data cannot be used to infer causality, therefore further investigation is required. Secondly, although the MoCA has proven useful in evaluating global cognitive function or screening measures for dementia, it is also influenced by an individual’s level of education, thus requiring varying cutoffs for individuals with different levels of education. This was controlled for in this study. Finally, frequency of exercise was obtained instead of exercise intensity, or type, etc. Additionally, only information about the quality of sleep was collected rather than its duration. Further research should be undertaken to explore how type and intensity of exercise and sleep duration work in the relationship between exercise, sleep, depression, and cognitive function.

## Figures and Tables

**Figure 1 ijerph-17-00709-f001:**
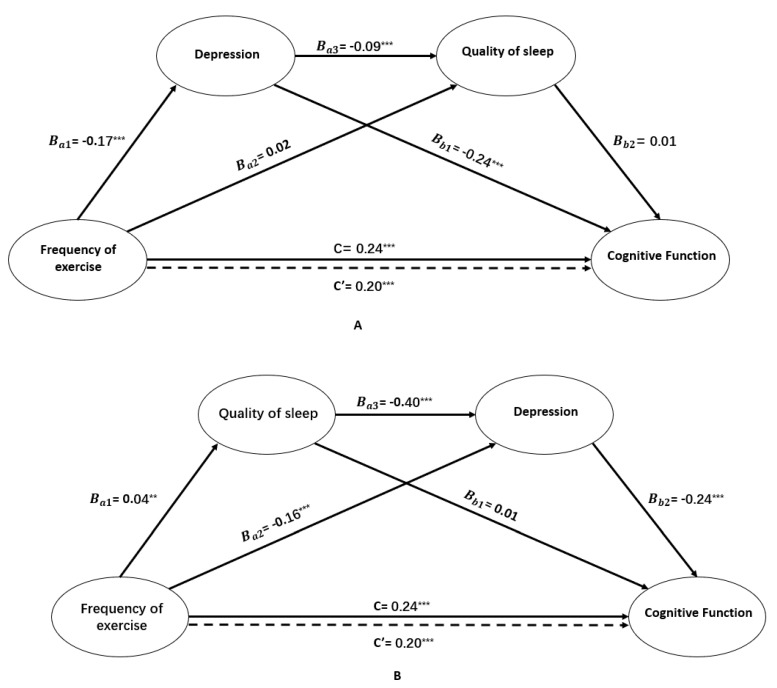
The serial multiple mediation role of depression and quality of sleep in the relationship between frequency of exercise and cognitive function and unstandardized beta values. c and c’ represent the total effect and direct effect of frequency of exercise on cognitive function, respectively. ^**^*p* < 0.01, ^***^*p* < 0.001. (**A**) presented serial-multiple mediation role of depression and quality of sleep in the relationship between frequency of exercise and cognitive function and unstandardized beta values for all participants; (**B**) provided serial-multiple mediation role of quality of sleep and depression in the relationship between frequency of exercise and cognitive function and unstandardized beta values.

**Table 1 ijerph-17-00709-t001:** Means (M), standard deviations (SD) and correlations among the variables.

Variable	Mean	SD	1	2	3	4
1. Frequency of Exercise	2.19	1.82	NA			
2. Quality of Sleep	2.39	1.05	0.08^***^	NA		
3. Depression	1.87	2.22	−0.16^***^	−0.21^***^	NA	
4. Cognitive Function	21.61	4.92	0.20^***^	0.08^***^	−0.21^***^	NA

Note:^***^*p* < 0.001. Depression was measured by the Geriatric Depression Scale (GDS), higher scores represented more severe depression. NA: not applicable.

**Table 2 ijerph-17-00709-t002:** Specific indirect effects of frequency of exercise through depression and quality of sleep on cognitive function.

			Bootstrapping	
Effect	Product of Coefficients		95% BCa Confidence Interval	
	Point Estimate	SE	Lower	Upper
Total Indirect Effect	0.1784	0.0543	0.1038	0.3158
FOE→DEP→CF	0.1762	0.0531	0.1034	0.3143
FOE→DEP→QOS→CF	0.0009	0.0055	−0.0097	0.0122
FOE→QOS→CF	0.0013	0.0084	−0.0125	0.0232

Note. FOE: frequency of exercise; DEP: depression; QOS: quality of sleep; CF: cognitive function.

**Table 3 ijerph-17-00709-t003:** Comparisons of indirect effects and specific indirect effects of frequency of exercise on cognitive function.

			Bootstrapping	
Effect	Product of Coefficients		95% BCa Confidence Interval	
	Point Estimate	SE	Lower	Upper
Total Indirect Effect	0.0428	0.0091	0.0265	0.0621
FOE→QOS→CF	0.0005	0.0030	-0.0053	0.0070
FOE→QOS→DEP→CF	0.0036	0.0014	0.0014	0.0069
FOE→DEP→CF	0.0387	0.0085	0.0235	0.0567
Specific Indirect Effect Contrast Definitions				
Path 1/Path 2	−0.0350	0.0082	−0.0525	−0.0205

Note. FOE: frequency of exercise; DEP: depression; QOS: quality of sleep; CF: cognitive function; Path 1: FOE→QOS→DEP→CF; Path 2: FOE→DEP→CF.

## Data Availability

The datasets generated and analyzed during the current study are not publicly available on the principle of confidentiality, but are available from the corresponding author on reasonable request.

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
