# Peer review of "Effect of Frequency of Exercise on Cognitive Function in Older Adults: Serial Mediation of Depression and Quality of Sleep"

_ijerph, 2020, doi:10.3390/ijerph17030709_

Round 1

Reviewer 1 Report

Dear authors,

It was a pleasure to read your manuscript which I found it pretty interesting. However, I have some questions that I would like you to answer. 

1.Is there any data on the type of exercise (e.g. resistance, aerobic, HIIT etc...) ? If yes, please report.

2.Is there any data on the intensity and duration of PA? If yes, please report.

3. I am just wondering how reliable it is to just subjectively measure the quality of sleep? I believe that the authors should expand on the confounding factors that possibly affect this.

4. No data has been reported with respect to the health status of participants. So, I would suggest including those data in the manuscript. 

5. Any difference in genders? please elaborate. 

6. What is the practical application of this study? This should be mentioned in the end of the manuscript.

Author Response

Dear Reviewer,

Thank you very much for your comments to our manuscript (ijerph-674837), and all are valuable and helpful for revising and improving the manuscript. Based on your comments and suggestions, we have made an extensive modification that we hope it can meet with approval. A revised manuscript with the correction sections yellow marked is attached. We addressed your concerns as outlined below:

Point 1: Is there any data on the type of exercise (e.g. resistance, aerobic, HIIT etc...)? If yes, please report.

Response 1: Thank you for your valuable comment. Yes, the type, intensity, and duration of exercise/PA were important to thorough understanding the association between exercise/PA and cognition. However, only frequency of exercise was asked, and of note the exercise in this study was limited to the physical activity with the purpose of improving health instead of work and housework and lasting more than 20 minutes (see page 3, line 95-97). The lack of type, intensity and duration of exercise/PA have been addressed as a limitation in discussion in the revised version (see page 7, line 61-64).

Point 2: Is there any data on the intensity and duration of PA? If yes, please report.

Response 2: Thank you for thoughtful suggestion. For this comment, please see our response to the Q1.

Point 3: I am just wondering how reliable it is to just subjectively measure the quality of sleep? I believe that the authors should expand on the confounding factors that possibly affect this.

Response 3: It is a good suggestion. In the revised version, the days of drinking tea per week and chronic diseases status (hypertension and diabetes) were added as possible confounders and we reconstructed the two multi-mediation models.

Point 4: No data has been reported with respect to the health status of participants.

Response 4: Thank you for your careful check. We have further disclosed the subjects’ self-reported health status and chronic diseases status (hypertension and diabetes) in the revised manuscript (see page4, line 139-141).                

Point 5: Any difference in genders? please elaborate.

Response 5: Thank you for your constructive comment. Per your suggestion, we ran several multi-intermediary models based on different genders and potential confounding variables like age, educational years, days of drinking tea per week were included. The results briefly presented as the following figures:

Figure A-C presented serial-multiple mediation role of “depression to quality of sleep” in the relationship between frequency of exercise and cognitive function and unstandardized beta values for all participants, men and women, respectively. As we can see, these tests revealed that no matter what the population would be, there was only one indirect path working (path: FOE-DEP-CF). Therefore, the results seemed consistent among males and females. Figure D-F provided serial-multiple mediation role of “quality of sleep to depression” in the relationship between frequency of exercise and cognitive function and unstandardized beta values for all participants, men and women, respectively. As we can see, further analyses showed that quality of sleep alone did not mediate the relationship mentioned above. Moreover, for all participants and women, frequency of exercise was associated with better quality of sleep, which in turn was associated with fewer depressive symptoms and finally contributing to better cognitive function. However, such results were not observed in men. Nevertheless, the coefficients between frequency of exercise and quality were quite small (For all participants, B=0.04; for men, B=0.03; for women, B=0.05), and considering the consistency of the full text, we did not to put this part of results in our revised manuscript.

Point 6: What is the practical application of this study? This should be mentioned in the end of the manuscript.

Response 6: Thanks for your good comments. We have added the following discussion on this issue in our revised manuscript, see page7, line 250-256.

Based on the general result of the current research, milder depression and better quality of sleep have been determined to function as a buffer against cognitive decline. In order to reduce seniors’ depressive symptoms and increase their quality of sleep and cognitive function, more physical activity programs like dancing and Baduanjing exercise could be recommended in communities. Taking a work, which is also considered as a convenient way to exercise, may be organized for different groups. And health education concerning benefits of physical exercise on depression, sleep quality, and cognitive function can be conducted to raise relevant issues.

Reviewer 2 Report

The article is very interesting because it addresses a growing problem, such as the population of older, more and more numerous. This paper analyzes the relationship between the frequency of exercise and cognitive function in older adults. There are some considerations on paper that could be improved: 1. The initials that appear in the summary make it difficult to read and understand it, since it is necessary to check what those initials refer to. 2. I think that it would be convenient to eliminate the word causal relationship that appears in several places in the text, since what they establish is a cross-sectional study of the relationship between variables of the models they propose.

Reviewer 3 Report

The authors present an interesting study on the mechanism of mediation between physical activity and cognitive function based on face-to-face survey. They found that depression and sleep are two mediators of this mechanism, and established multi-mediation models which include physical activity, quality of sleep, depression, and cognitive function. The structure of this paper is fine, and the analysis of research data is well organized. Therefore, I recommend revisions as follows:

Figure 1 demonstrate the findings of two tested models, while it is not clear what are the differences between these two models? Why to compare these two models? And what are the meanings of Ba1, Bb2…? Pls add the explanations for Figure 1A and Figure 1B in the caption of Figure 1. Pls adjust Table 3 to fit in one page Line 93, “Frequency of exercise was measured with the question…” is recommended to be modified as “Frequency of exercise was obtained …” since the FOE is not “measured”.

Round 2

Reviewer 1 Report

Dear authors,

Thanks so much for taking my comments into consideration and revising the manuscript accordingly. At this point, I have no further comments to add.